# ToolTalk: Evaluating Tool Usage in a Conversational Setting

## Abstract

Large language models (LLMs) have displayed massive improvements in reasoning and decision-making skills and can hold natural conversations with users. Many recent works seek to augment LLM-based assistants with external tools so they can access private or up-to-date information and carry out actions on behalf of users. To better measure the performance of these assistants, this paper introduces ToolTalk, a benchmark consisting of complex user intents requiring multi-step tool usage specified through dialogue. ToolTalk contains 28 tools grouped into 7 plugins, and includes a complete simulated implementation of each tool, allowing for fully automated evaluation of assistants that rely on execution feedback. ToolTalk also emphasizes tools that externally affect the world rather than only tools for referencing or searching information. We evaluate GPT-3.5 and GPT-4 on ToolTalk resulting in success rates of 26% and 50% respectively. Our analysis of the errors reveals three major categories and suggests some future directions for improvement.

## 1 Introduction

Large language models (LLMs) can perform impressive feats in natural language understanding, generation, and other tasks involving manipulation of text. With appropriate adjustments after pre-training, they can hold fluent and natural conversations with users. However, the scope of such conversations is still limited by LLMs lacking access to knowledge outside of their training data, exhibiting limited mathematical reasoning and computational abilities, and otherwise being unable to interact with the outside world.

To overcome these limitations, various prior works have proposed integrating LLM-powered chatbots with the ability to use tools such as search engines (Nakano et al., 2022), calculators, or web APIs (Mialon et al., 2023). Making meaningful progress in tool use requires relevant benchmarks and evaluation datasets that can fully exercise these systems with realistic and challenging conversations. In this paper, we introduce ToolTalk as a step towards this goal. ToolTalk consists of 78 conversations with 178 total turns, making use of 28 unique tools grouped into 7 categories, along with an evaluation methodology tailored towards measuring accurate tool use.

Several considerations informed our design of ToolTalk in order to best simulate typical conversations that a user may wish to have with an LLM-based assistant. First, we wanted to ensure that ToolTalk is *conversational*, and allows for multiple rounds of dialogue between the user and the assistant for a single intent; reflecting how users may not always wish to formulate their full request in one utterance and can add additional qualifiers or issue corrections after receiving some feedback from the assistant. This allows us to include user intents requiring a complex series of tool invocations without having unnaturally long utterances. Second, we include a ground-truth set of tool calls that should have been made for each user utterance, suitable for use in an automated evaluation comparing against the tool calls predicted by an assistant. Third, ToolTalk includes executable implementations of every tool included in the dataset, to facilitate the evaluation of assistants that may consider results from prior tool invocations to decide which ones to make next. Fourth, ToolTalk includes tools intended to have side effects (such as sending emails, or adding/deleting calendar events), which we refer to as "action tools", rather than only making database queries (such as searching for emails containing a particular keyword). Such action tools are necessary if the assistant is to automate the user's tasks.

We tailor our evaluation methodology towards the particulars of our dataset design, going beyond common metrics like exact-match accuracy. In particular, we separately consider invocations of action and non-action tools, considering that incorrect invocations to action tools, such as sending a message to the wrong person, may have particularly negative effects for the user. On the other hand, if the assistant makes both correct non-action tool invocations and some incorrect extraneous ones, the extraneous ones may still provide useful information to the user (even if it's not what the user directly requested). As such, we use *tool invocation recall* and *incorrect action rate* as the primary metrics within a single conversational turn, and define a conversation-level notion of *success*.

We apply ToolTalk on two assistants implemented using the function calling support of OpenAI's Chat completions API with the GPT-3.5 and GPT-4 models. We found that `gpt-3.5-turbo-0613` and `gpt-4-0613` achieve a conversation-level success rate of 26% and 50% respectively, demonstrating that tool usage in a conversational setting is still a difficult task for even some of the most state-of-the-art models. We then conduct further analyses to determine reasons why GPT-3.5 and GPT-4 fail on conversations. We find that both GPT-3.5 and GPT-4 can hallucinate arguments, fail to understand documentation, and even outright claim to have accomplished a task without calling any tools.

Our paper makes the following contributions:

- We introduce a conversational dataset for tool-using LLM-powered assistants, containing a broad range of tools and example conversations with ground truth annotations for tool invocations that allow for an automated evaluation.
- We ensure that the dataset contains multi-turn conversations requiring use of multiple tools, including tools with side effects, to better simulate how users may interact with a tool-using assistant.
- We develop an evaluation methodology which reflects the differences between tools with side effects and tools without them.
- We evaluate assistants built using GPT-3.5 and GPT-4 using our dataset and analyze their errors, finding issues such as hallucinated arguments and misunderstood documentation.

## 2 DATASET DESIGN

### 2.1 PLUGINS AND TOOLS

ToolTalk is designed for a paradigm where individual users will be able to customize a personal assistant with a number of *plugins* available through various online stores. This can be seen as similar to how a user might customize their phone with apps of various functionality. Each plugin contains a set of *tools* designed around a single purpose such as managing a calendar, buying movie tickets, or listening to music. We define a tool as a single function needed to accomplish that purpose such as creating an event, searching for movies, or playing a song. We assume that most plugins will need to contain multiple tools. For example, a theoretical "Calendar" plugin should not only have the ability to create events, but also to then search, modify, and delete these events.

For our dataset, we defined 7 plugins containing a total of 28 tools (see Appendix A for the full list). Using similar domains as those in Li et al. (2023), we created the following plugins:

- **AccountTools**: containing tools for account management such as logging in and out, updating account information, or looking up other users.
- **Alarm**: adding, deleting, and finding alarms.
- **Calendar**: creating, modifying, deleting, and searching events and meetings
- **Email**: searching inbox and sending emails
- **Message**: sending and reading messages from other users
- **Reminder**: setting, completing, and deleting reminders on a to do list
- **Weather**: querying current weather, weather forecasts, and historic weather data based on location

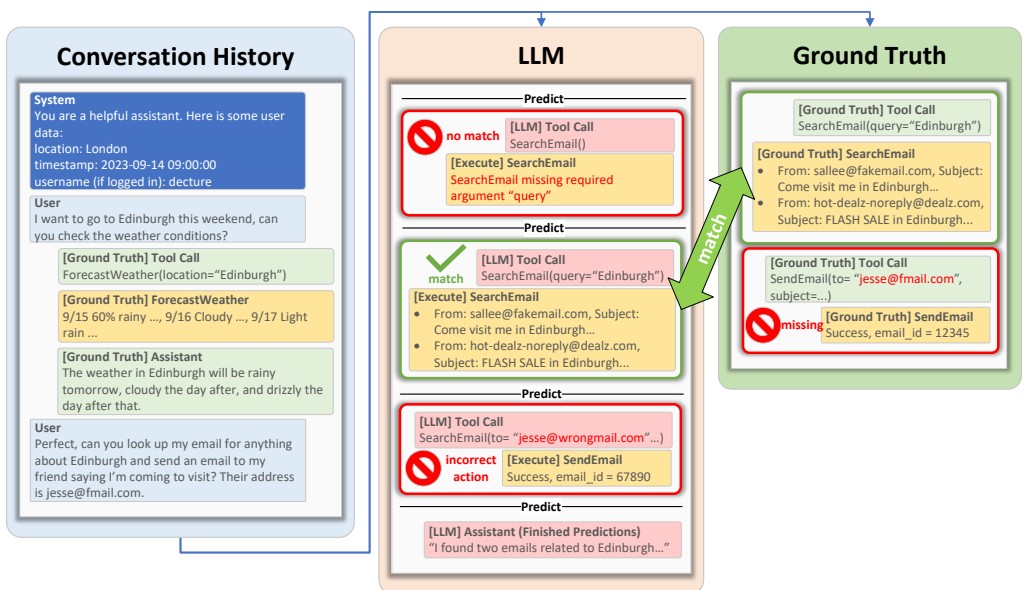

Figure 1: ToolTalk methodology. A system prompt, user and assistance utterances, and ground truth tool calls are fed as conversation history to the LLM. We prompt the LLM for a tool call prediction and simulate execution. This is added to the conversation history and the LLM is prompted for another prediction. This continues until the LLM predicts an assistant response. LLM predictions are then forgotten and the process is repeated for the next assistant turn. Predicted tool calls are then compared against ground truth tool calls.

To teach the LLM about how to use the tools, each tool contains a high-level description, verbose documentation about each of its parameters, and a description of its return value. To facilitate evaluation, each tool has a simulated implementation in Python, along with a method to judge whether two invocations of the same tool with different parameters should be considered equivalent. We also note for each tool whether it is considered an action (has side effects) or not. We also include accompanying databases with mock information about fictional existing users, emails, reminders, and so on, for the simulated tool implementations to use.

## 2.2 CREATING CONVERSATIONS

To help create realistic conversations that exercise our tools and plugins, we used GPT-4. For each subset of 3 plugins from the 7 plugins we have defined, we create prompts which lists the documentation for all the tools in these 3 plugins, and instructs GPT-4 to create 3 realistic scenarios involving a user trying to accomplish a task that uses at least 5 tool calls from the random subset of plugins. We create as many prompts as the number of tools that exist in the subset of 3 plugins currently under consideration, such that each prompt instructs GPT-4 to specifically use one of the tools in the subset of 3 plugins. We provide the prompt template used in Appendix B.

The above procedure results in the generation of ~400 scenarios. We then repeatedly sampled a scenario evenly from all tools, discarding sampled scenarios that do not involve the required tool, hallucinate non-existent tools, or seem implausible. Using a sampled scenario as general guidance, we manually create a conversation, writing down all of its parts by hand.

Each conversation consists of a user utterance, the tool calls that the assistant should make given that utterance, the expected return values for those calls, and the assistant's natural language responses given the user's utterances plus the tool calls and their results, repeating in that order until the conversation is finished. As metadata for the conversation, we also specified a timestamp for the

conversation, and the user's location and username.[1] We ensure that each conversation contains at least 3 tool calls. We repeat the above sampling of scenarios until we have written 50 conversations.

Additionally, we create 28 "easy" conversations completely by hand, one for each tool. This easy version of ToolTalk consists of a few turns of user-assistant dialogue followed by a single tool call. Combined with the prior 50 "hard" examples, we create a total of 78 conversations comprising ToolTalk.

After constructing conversations, we ensure that the databases used by our simulated tool implementations contain the necessary content so that when we execute the ground truth tool calls as listed in the conversations we have created, they return the same ground truth values.

## 3 EVALUATION METHODOLOGY

Evaluation of a tool-using assistant with ToolTalk consists of two phases. In the first phase, for each conversation, we take all prefixes that end in a user utterance (which could have been preceded by prior user utterances, the tool calls made for those utterances, the results of those calls, and the assistant's response considering all of the above). We run the assistant with this prefix, where it can either predict a tool call or generate a response given the calls already made and their results; if the assistant predicts a tool call, we execute it using our simulated tool implementations and then provide the assistant with the result. In the second phase, for each conversation prefix, we compare the tool calls predicted for that prefix against its corresponding ground truth, computing the *tool invocation recall* and *incorrect action rate* as described below.

### 3.1 TOOL CALL CORRECTNESS

As described in Section 2.1, for each action tool, we defined a function to compare a predicted and a ground truth invocation of that tool (considering the arguments in the invocations), to help us determine whether a predicted tool call should be considered equivalent to one in the ground truth. For example, if an email is required to be sent to multiple people, we only check that the set of emails are the same instead of requiring the exact same order.

For argument fields that accept free-form natural language inputs, such as message bodies and event descriptions, we compute their embeddings with DistilBERT using `sent2vec`[2] and check whether their cosine similarity is above 0.9.

For optional arguments, if the ground truth invocation has a value for one, then we compare its value against the one in the predicted invocation; if the ground truth invocation is missing a value for an optional argument, then it is entirely disregarded and the predicted call may have any value for that argument (or none at all) while still being considered correct. For example, the description of a calendar event is an optional argument, and if it is not explicitly mentioned in the conversation, then it is unlikely to impact the correctness of a predicted call whether or not it is filled out.

For the non-action tools (which are generally tools for searching over a database), we do not compare the arguments in the tool calls, but rather compare the execution results of the predicted and ground truth tool calls. They are considered equivalent of the results are identical.

### 3.2 CONVERSATION SIMULATION

Algorithm 1 shows the general pseudocode for conversation simulation. To simulate a conversation, we first reset the state of the world (e.g. databases get reset to their initial state). For each turn in the ground truth (consisting of a user's utterance, tool calls for that utterance, and the assistant's reply), we provide the information from all previous turns, followed by the current turn's user utterance, to the model. We then let the model predict as many tool calls as it wants, executing them one at a time until the prediction model produces a reply to the user instead of a tool call.

---

[1]For scenarios that use tools such as UserLogin or RegisterAccount, we omit the username to simulate a user that has yet to log in or have an account.

[2]https://github.com/pdrm83/sent2vec

**Algorithm 1** Conversation simulation

---

**Require:** conversation $T$ an array of turns
**Require:** Each turn contains a user utterance, ground truth tool calls, and a ground truth assistant reply
**Require:** tool prediction function $LLM$
**Require:** tool execution function $Exec$
1:   $h \leftarrow []$ # conversation history
2:   $p \leftarrow []$ # predictions
3:   **for** $t \in T$ **do**
4:      $h$.append($t$.user_utterance)
5:      $u \leftarrow []$ # turn history
6:      $c \leftarrow LLM(h + u)$ # current prediction
7:      **while** $c$ is not assistant reply **do**
8:         $c$.exec_feedback $\leftarrow Exec(r)$
9:         $u$.append($c$)
10:        $p$.append($c$)
11:        $c \leftarrow LLM(h + u)$
12:      **end while**
13:      $h$.extend($t$.ground_truth_tools)
14:      $h$.append($t$.ground_truth_assistant_reply)
15: **end for**
16: **return** $p$

**Algorithm 2** ToolTalk evaluation

---

**Require:** tool predictions for single conversation $P$
**Require:** ground truth tool calls for single conversation $G$
1:   $M \leftarrow \emptyset$ # matches
2:   **for** $g \in G$ **do**
3:      **for** $p \in P$ **do**
4:         **if** $g$.match($p$) **then**
5:           $M \leftarrow M \cup \{p\}$
6:           break
7:         **end if**
8:      **end for**
9:   **end for**
10: $A \leftarrow \forall p \in P$ where $p$ is action
11: $I \leftarrow \forall a \in A$ where $a \notin M$
12: precision $\leftarrow |M|/|P|$
13: recall $\leftarrow |M|/|G|$
14: incorrect action rate $\leftarrow |I|/|A|$
15: success $\leftarrow (M == G) \wedge (I == \emptyset)$
16: **return** precision, recall, incorrect action rate, success

### 3.3 INCORRECT ACTIONS

Each tool is labeled as being either an action or not. We consider a tool an action if its execution has the ability to affect the external world such as sending messages or deleting calendar events. In comparison, non-action tools only passively references knowledge from the outside world such as looking up the weather or calling a calculator. We make this distinction between action and non-action tools because incorrect calls to action tools are much more consequential. For example, an incorrect call to the DeleteAlarm tool could result in the user over-sleeping. While an assistant could theoretically realize that it made an incorrect action tool call and make a different one to reverse its effects, not all actions are reversible.

Thus, during evaluation, we also track "incorrect" actions. We consider an action "incorrect" if the tool called is labeled as an action, it fails to match any call in the ground truth, and if the tool call executed without any errors (including by having the correct number of arguments and passing the correct types).[3]

### 3.4 METRICS

$$\forall g \in G; g \in M \iff \exists p \in P \text{ where } f_{tool}(p, g) \tag{1}$$

$$\text{success} = (M == G) \wedge (I == \emptyset) \tag{2}$$

We use the tool call correctness function, $f_{tool}$, to compare each prediction to all tool calls in the ground truth; as described in Algorithm 2, each ground truth tool call can only match once to a predicted tool call. Given a set of $M$ predictions matching ground truth (defined in equation 1), the set of all predictions $P$, and the set of all ground truth tool calls $G$ we calculate precision and recall as $|M|/|P|$ and $|M|/|G|$ respectively. Additionally, we define $A$ as the set of all actions predicted and $I$ as the set of incorrect actions and calculate incorrect action rate as $|I|/|A|$.

Additionally, we compute success as a boolean value for each conversation, following Equation 2. The assistant succeeds at a conversation if and only if it has perfect recall and no incorrect actions.

---

[3]For the SendEmail and SendMessage tools, we ignore errors which occur due to invalid recipient emails or usernames.

| Model | Subset | Success rate | Precision | Recall | Incorrect action rate |
|-------|--------|-------------|-----------|--------|----------------------|
| GPT-3.5 | Easy | 85.7% | 42.4% | 89.3% | 5.0% |
| GPT-4 | Easy | 92.8% | 69.2% | 96.4% | 3.8% |
| GPT-3.5 | Hard | 26.0% | 54.6% | 69.7% | 23.9% |
| GPT-4 | Hard | 50.0% | 74.9% | 79.0% | 25.1% |

Table 1: GPT-3.5 and GPT-4 evaluated on easy and hard versions of ToolTalk.

We take success rate over all conversations as our key metric. Since success rate is a composite of two scores, we keep recall and incorrect action rate as additional metrics to provide more detail. We also include precision as a measure of efficiency in tool prediction; a higher precision indicates that there were fewer predicted tool calls that are unnecessary according to the ground truth.

## 4 EXPERIMENTS AND ANALYSIS

### 4.1 EXPERIMENTS

We evaluate GPT-3.5 (`gpt-3.5-turbo-0613`) and GPT-4 (`gpt-4-0613`) on ToolTalk using the functions functionality as part of OpenAI's Chat completions API (OpenAI). This API takes as input an optional system message, a history of messages between a user and an assistant, tool documentation, and any previous tool invocations and their responses, and produces as output either a tool invocation or an assistant message.

In the system message, we include the conversation's location, timestamp, and (if present) username. We supply documentation for all 28 tools at once to simulate a user with all 7 plugins enabled. We then simulate and evaluate all conversations in the easy and hard subsets of ToolTalk, following Algorithms 1 and 2.

Table 1 shows the results. We get success rates of 85.7% and 92.8% for GPT-3.5 and GPT-4 on the easy version of ToolTalk, and success rates of 26.0% and 50.0% on the hard version. GPT-4 outperforms GPT-3.5, but still achieves similar incorrect action rates. From precision, we can see that GPT-4 is also more efficient than GPT-3.5. However, performance for both models are low, showing the difficulty of tool usage in conversation settings.

### 4.2 ANALYSIS

We analyze the conversations that either GPT-4 or GPT-3.5 fail on. We notice that for both LLMs, there are three major reasons that they can fail. First, the model may predict a tool call prematurely on a turn before a user has provided the necessary information. Second, the model may exhibit poor planning, resulting in omitting or using the wrong tools. Third, it may have picked the correct tool to use, but invoked it with incorrect or missing arguments, failing to follow the tool's function signature described in the documentation. GPT-3.5 is more susceptible to these errors, but they manifest as well for GPT-4.

**Premature tool calls.**   This usually occurs when the user has a clear intent, e.g. "I want to create an event", but has yet to provide the necessary information to provide as arguments. It then manifests as hallucinating plausible values to supply as arguments. This is harmless when predicting reference tools but is a direct contribution to failure when predicting action tools. Concerningly, even when the hallucinated arguments will result in execution errors, the model will persist in hallucinating more arguments. Despite these issues, both GPT-3.5 and GPT-4 will generally choose the correct tools to accomplish the intent.

**Faulty reasoning.**   Ultimately, premature tool calls could be mostly explained by faulty reasoning, where the LLM fails to reflect that it does not have all the information it needs to accomplish a task and needs to ask the user to provide more clarification. Similarly, omission or the usage of wrong tools can also be explained by faulty reasoning skills; rather than reflecting and realizing it needs

| Model | Premature tool calls | Faulty planning | Incorrect tool invocations | Total failures |
|-------|---------------------|-----------------|----------------------------|----------------|
| GPT-3.5 | 26.9% | 53.7% | 19.4% | 67 |
| GPT-4 | 32.0% | 42.0% | 26.0% | 50 |

Table 2: Percent of failing error types out of all failing turns for GPT-3.5 and GPT-4.

to ask the user to provide more clarification, the LLM fails to realize that it needs to call additional tools in order to accomplish a task.

For example, the SendEmail tool requires a recipient email address, which can be obtained from a username with the QueryUser tool. However, instead of using QueryUser and then passing its result to SendEmail, the model may instead hallucinate a plausible email address belonging the user. In other circumstances, the model will forget specifics of the task and fail to call the corresponding tools. For example, if a user wants to both send a message and change their calendar, the model will only change the calendar and not send the message. In egregious cases, both LLMs can hallucinate tools or not predict any tool usage at all and confidently state that it has accomplished the task.

**Incorrect invocations of the correct tool.** Even if the model picks the correct tool, it can invoke the tool with incorrect arguments, by missing values or supplying wrong values. This can happen from failing to understand documentation, failing to understand the output of previous tool invocations, or weak mathematical skills. Examples include supplying 2 PM as "2:00" instead of "14:00"; calculating a 10 hour event ending at 6 PM as 6 PM to 12 AM; incorrectly supplying a reminder it had just created to the DeleteReminder tool.

**Quantitative results.** Table 2 shows the number of turns in which the above error types occur, in our evaluation of GPT-4 and GPT-3.5. We determine error types automatically by comparing predictions for a single turn with the ground truth for the same turn and seeing which predictions and ground truth tool calls fail to find a match. GPT-4 overall produces fewer errors for each category than GPT-3.5. However, GPT-4 generally fails for the same reasons as GPT-3.5 in cases where both fail on the same conversation. GPT-4 does demonstrate a clear improvement in planning over GPT-3.5 as GPT-4 will generally be able to determine all tools needed to accomplish a task.

**Lessons.** Our results and analyses suggest a few ways to improve tool usage and design for LLMs. Some form of self-reflection or grounding for argument values seems key to reduce premature invocation of tools. This can also help LLMs determine if it has all the tools necessary to complete a task. For GPT-3.5 in particular, minimizing the number of arguments in tools seems likely to lead to good improvements. This is because unlike GPT-4, GPT-3.5 has more difficulty recovering from errors, often giving up.

### 4.3 EXPERIMENT REMOVING DOCUMENTATION

We perform an ablation study to measure the effect of tool documentation by removing all tool and parameter descriptions keeping only names and parameter types. We re-evaluate GPT-3.5 and GPT-4 on ToolTalk producing Table 3. We also re-run our analysis on error types producing Table 4. Performance on ToolTalk significantly decreases across the board except for incorrect action rate. The decrease in incorrect action rate could be due to tools being harder to use, resulting in less successful tool executions overall, whether or not it matches ground truth.

From Table 4 we can see that faulty planning accounts for the majority of errors produced by GPT-3.5 and GPT-4. We perform a qualitative analysis and discover both models tend to call tools with incorrectly formatted arguments, receive errors in execution feedback, then persist in the same incorrect format. This results in both models eventually giving up and predicting an assistant reply thereby missing all other tool calls in the ground truth.

| Model | Subset | Success rate | Precision | Recall | Incorrect action rate |
|-------|--------|--------------|-----------|--------|-----------------------|
| GPT-3.5 | Easy | 82.1% | 35.8% | 85.7% | 2.2% |
| GPT-4 | Easy | 85.7% | 52.0% | 92.9% | 5.7% |
| GPT-3.5 | Hard | 16.0% | 40.1% | 62.6% | 11.8% |
| GPT-4 | Hard | 34.0% | 40.6% | 64.3% | 13.0% |

Table 3: GPT-3.5 and GPT-4 evaluated **without documentation** on ToolTalk.

| Model | Premature tool calls | Faulty planning | Incorrect tool invocations | Total failures |
|-------|----------------------|-----------------|----------------------------|----------------|
| GPT-3.5 | 12.3% | 71.2% | 16.4% | 73 |
| GPT-4 | 16.2% | 60.3% | 23.5% | 68 |

Table 4: Error types **without documentation** for GPT-3.5 and GPT-4.

# 5 RELATED WORK

In Section 1, we described our desired criteria for evaluating tool-using LLM-based assistants: using *dialogue* to specify intents requiring *multi-step* tool invocations, and *actions* rather than only retrieving information, for a fully *automated* evaluation not requiring human judgement over the outputs of the system under test. Table 5 summarizes how other work about evaluating tool-using LLMs compares along these factors. We describe the related work in greater detail below.

Tool-augmented LLMs are also known as tool-augmented learning, tool LLMs, tool-learning, augmented language models (ALMs), or tool manipulation with LLMs (Xu et al., 2023; Mialon et al., 2023; Qin et al., 2023a). Development in this area consists of improving LLM performance in traditional tasks by giving them access to tools such as a calculator or search engine (Lu et al., 2023a; Yao et al., 2022b; Paranjape et al., 2023; Hao et al., 2023). It can also include applying LLMs to traditional automation tasks such as embodied robotics or browsing the web (Liu et al., 2023b; Deng et al., 2023; Yao et al., 2022a; Liang et al., 2023), dubbed "LLM-as-agent" by AgentBench (Liu et al., 2023a).

Traditional tasks that tool-augmented LLMs have been applied to include question answering such as ScienceQA (Saikh et al., 2022) or HotPotQA (Yang et al., 2018), mathematical reasoning (Cobbe et al., 2021; Lu et al., 2023b; Qiao et al., 2023), multilingual translation and QA (Lewis et al., 2020; Scarton et al., 2019), open-domain QA (Zhu et al., 2021), and commonsense QA (Talmor et al., 2019) to name a few. These tasks are useful for demonstrating the benefits of augmenting LLMs with tool usage, but fail to fully distinguish how much LLMs rely on internal knowledge vs good usage of tools (Zhuang et al., 2023). They also fail to incorporate the use of tools that affect the external world since they are unnecessary for those tasks.

Common agent benchmarks that have been applied to tool-augmented LLMs include WebShop (Yao et al., 2022a), Tabletop (Liang et al., 2023), Mind2Web (Deng et al., 2023), and ALFWorld (Shridhar et al., 2020). Additionally, AgentBench compiles Mind2Web, WebShop, and ALFWorld into a unified benchmark while adding additional agent environments such as interacting with a bash terminal, creating SQL commands to query a database, interacting with a knowledge graph, digital card game simulations, and lateral thinking puzzles (Liu et al., 2023a). ToolBench does something similar by compiling Tabletop and Webshop while introducing a variety of other tasks consisting of predicting a single API call. These benchmarks are useful for evaluating the effectiveness of tool-augmented LLMs in a variety of autonomous situations. However, none of them test tool-augmented LLMs in a conversational setting. Furthermore, tasks in these benchmarks consist of issuing a single utterance which an agent then tries to accomplish without any further human interaction. This is in contrast to ToolTalk, where a conversation will consist of multiple utterances with multiple intermediary tasks.

Past works have also created datasets for evaluating tool-augmented LLM-based assistants. Examples include ToolLLM (Qin et al., 2023b), API-Bank (Li et al., 2023), TPTU (Ruan et al., 2023), Gorilla (Patil et al., 2023), RestGPT (Song et al., 2023), GPT4Tools (Yang et al., 2023), and ToolAlpaca (Tang et al., 2023) among others. Unfortunately, many of these datasets require manual inspec-

| | No. of tools | Dialogue | Complex | Actions | Automated |
|---|---|---|---|---|---|
| ReAct (Yao et al., 2022b) | 3 | ✗ | ✗* | ✗ | ✓ |
| ART (Paranjape et al., 2023) | 3 | ✗ | ✗* | ✗ | ✓ |
| Tool Learning (Qin et al., 2023a) | 17 | ✗ | ✓ | ✓ | ✓ |
| Toolformer (Schick et al., 2023) | 5 | ✗ | ✓ | ✗ | ✓ |
| Chameleon (Lu et al., 2023a) | 15 | ✗ | ✓ | ✗ | ✓ |
| ToolkenGPT (Hao et al., 2023) | 58 | ✗ | ✓ | ✓ | ✓ |
| ToolQA (Zhuang et al., 2023) | 13 | ✗ | ✓ | ✗ | ✓ |
| API-Bank (Li et al., 2023) | 53 | ✓ | ✗* | ✓ | ✓* |
| ToolBench (Xu et al., 2023) | 232 | ✗ | ✗* | ✓ | ✓ |
| AgentBench (Liu et al., 2023a) | 100+ | ✗ | ✓ | ✓ | ✓ |
| TPTU (Ruan et al., 2023) | 12 | ✗ | ✗ | ✓ | ✓ |
| Gorilla (Patil et al., 2023) | 1,645 | ✗ | ✗ | ✓ | ✓ |
| RestGPT (Song et al., 2023) | 94 | ✗ | ✓ | ✓ | ✗ |
| GPT4Tools (Yang et al., 2023) | 31 | ✗ | ✗ | ✓ | ✓ |
| ToolLLM (Qin et al., 2023b) | 16,464 | ✗ | ✓ | ✓ | ✓ |
| ToolAlpaca (Tang et al., 2023) | 400 | ✗ | ✓ | ✓ | ✗ |
| **ToolTalk** | 28 | ✓ | ✓ | ✓ | ✓ |

Table 5: Comparison of evaluation used in prior work with ToolTalk. We note total number of tools used (No. of tools), if any task is specified over multiple user utterances (dialogue), if any task requires more than 1-2 tools to complete (complex), if any task requires the use of action tools (actions), and if all evaluation is done automatically (automated). We note nuances in prior work denoted by "*" in Appendix D.

tion of the outputs of the assistant under test to perform a complete evaluation. A lot of them also have unrealistic queries, and do not reflect questions or intents humans are likely to say in real life.[4] Many of them are also simple, where the solution requires one or two tool calls (Li et al., 2023; Ruan et al., 2023; Yang et al., 2023; Tang et al., 2023). Except for Li et al. (2023), these consider users' utterances in isolation rather than as part of a conversation or dialogue.

There also exists a corpus of work on task-oriented dialogue systems. This area of research is focused on collecting realistic, task-oriented dialogue for the tasks of intent classification and slot filling (Larson & Leach, 2022). Some popular task-oriented dialogue datasets include MultiWoz (Budzianowski et al., 2018), Taskmaster and TicketTalk (Byrne et al., 2019; 2020), and STAR and STARv2 (Mosig et al., 2020; Zhao et al., 2022). The goals of creating realistic dialogue and evaluating on intent classification and slot filling have some overlap with ToolTalk. However, task-oriented dialogue datasets usually only predict a single intent per user utterance, do not simulate plugins or tools, and do not provide execution feedback for predicted tool calls. TicketTalk (Byrne et al., 2020) is notable in that it does provide a simulation of a movie booking API, however this API does not provide execution feedback and is not rigorously defined allowing for loose arguments like "here" or "now".

## 6 CONCLUSION

We present ToolTalk, a new benchmark for evaluating tool-augmented LLMs in a conversational setting. Our benchmark emphasizes complex orchestration of multiple tools in a conversational setting. We provide simulated implementations of all tools, allowing for a fully automated evaluation where the LLM can decide which tools to further invoke based on the results of prior tool calls. Finally, we also introduce a unique form of evaluating correctness that takes into account unique aspects of individual tools and whether a tool usage system produces incorrect actions. We evaluate GPT-3.5 and GPT-4 using our dataset and methodology and analyze their errors, finding three major categories: premature tool calls, faulty reasoning, and incorrect invocations of the correct tool. In the future, we hope to expand the scope of this dataset to more conversations and simulate even more, diverse plugins.

---

[4]We include a few examples from various papers in Appendix C.

## 7 REPRODUCIBILITY

We will release ToolTalk shortly after getting the requisite approval. We include the exact versions of GPT-3.5 (`gpt-3.5-turbo-0613`) and GPT-4 (`gpt-4-0613`) available through the OpenAI API to be able to reproduce our results after release. We include the prompt used to generate our scenarios in Appendix B. We include information on system prompts and our application of OpenAI's Chat completions API in Section 4.1.

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

## A COMPLETE LIST OF TOOLS

We include the complete list of plugins and tools used in ToolTalk, and their corresponding descriptions.

**AccountTools** This API contains tools for account management.

- **ChangePassword** Changes the password of an account.
- **DeleteAccount** Deletes a user's account, requires user to be logged in.
- **GetAccountInformation** Retrieves account information of logged in user.
- **LogoutUser** Logs user out.
- **QueryUser** Finds users given a username or email.
- **RegisterUser** Register a new user.
- **ResetPassword** Resets the password of a user using a verification code.
- **SendVerificationCode** Initiates a password reset for a user by sending a verification code to a backup email.
- **UpdateAccountInformation** Updates account information of a user.
- **UserLogin** Logs in a user.

**Alarm** This API contains tools for managing alarms.

- **AddAlarm** Sets an alarm for a specific time.
- **DeleteAlarm** Removes an alarm given an alarm id.
- **FindAlarms** Finds alarms a user has set.

**Calendar** This API lets a users manage events in their calendar.

- **CreateEvent** Adds events to a user's calendar.
- **DeleteEvent** Deletes events from a user's calendar.
- **ModifyEvent** Allows modification of an existing event.
- **QueryCalendar** Queries for events that occur in a time range.

**Email** This API lets a user search and send emails.

- **SearchInbox** Searches for emails matching filters returning 5 most recent results.
- **SendEmail** Sends an email on behalf of a given user.

**Message** This API lets a user search and send messages.

- **SearchMessages** Searches messages matching filters returning 5 most recent results.
- **SendMessage** Sends a message to another user.

**Reminder** A suite of APIs for managing reminders for a TODO list.

- **AddReminder** Add a reminder with an optional due date.
- **CompleteReminder** Complete a reminder.
- **DeleteReminder** Delete a reminder.
- **GetReminders** Get a list of reminders.

**Weather** Get weather information of a location.

- **CurrentWeather** Get the current weather of a location.
- **ForecastWeather** Get the 3-day weather forecast of a location.
- **HistoricWeather** Get historic weather information of a location by month.

## B SCENARIO PROMPT

```
# Task
You will be provided with a list of APIs. These APIs will have a
description and a list of parameters and return types for each tool. Your
 task involves creating 3 varied, complex, and detailed user scenarios
that require at least 5 API calls to complete involving at least 3
different APIs. One of these APIs will be explicitly provided and the
other two will be chosen by you.

For instance, given the APIs: SearchHotels, BookHotel, CancelBooking,
GetNFLNews. Given that GetNFLNews is explicitly provided, your scenario
should articulate something akin to:

"The user wants to see if the Broncos won their last game (GetNFLNews).
They then want to see if that qualifies them for the playoffs and who
they will be playing against (GetNFLNews). The Broncos did make it into
the playoffs, so the user wants watch the game in person. They want to
look for hotels where the playoffs are occurring (GetNBANews +
SearchHotels). After looking at the options, the user chooses to book a
3-day stay at the cheapest 4-star option (BookHotel)."

This scenario exemplifies a scenario using 5 API calls. The scenario is
complex, detailed, and concise as desired. The scenario also includes two
 APIs used in tandem, the required API, GetNBANews to search for the
playoffs location and SearchHotels to find hotels based on the returned
location. Usage of multiple APIs in tandem is highly desirable and will
receive a higher score. Ideally each scenario should contain one or more
instances of multiple APIs being used in tandem.
```

```
Note that this scenario does not use all the APIs given and re-uses the "
GetNBANews" API. Re-using APIs is allowed, but each scenario should
involve at least 3 different APIs. Note that API usage is also included
in the scenario, but exact parameters are not necessary. You must use a
different combination of APIs for each scenario. All APIs must be used in
 at least one scenario. You can only use the APIs provided in the APIs
section.

Note that API calls are not explicitly mentioned and their uses are
included in parentheses. This behaviour should be mimicked in your
response.

Deliver your response in this format:
```
- Scenario 1: <Scenario1>
- Scenario 2: <Scenario2>
- Scenario 3: <Scenario3>
```

# APIs
```
{{API_DOCS}}
```

# Response
Required API: {{REQUIRED_API}}
Scenarios with >=5 API calls:

```
- Scenario 1:
```

## C  UNREALISTIC QUERIES

Below are some examples of unrealistic queries gathered from various sources. These queries are useful for generating potentially complex tool interactions or unusual combinations of tools. However, as a consequence, they are unrealistic for various reasons such as forcing the usage of disparate APIs in situations a human is unlikely to ask for, explicitly asking for API endpoints which end users are unlikely to know of, or generally being unnaturally long and explicit.

- "I'm working on a logistics project for my company and need to check the health of the SQUAKE API. Can you verify the API health by calling the 'Checkhealth' API endpoint? Additionally, I would like to retrieve the list of projects using the 'Projects' API endpoint." (Qin et al., 2023b)

- "How many singers have the average number of albums of singers in Beijing? Gives the square root of this number." (Ruan et al., 2023)

- "I am looking for x-large, red color women faux fur lined winter warm jacket coat, and price lower than 70.00 dollars." Yao et al. (2022a)

- "Can you retrieve the contact details of the 'Gondrand' customs agency in New Caledonia? I'm particularly interested in their postal code, email, name, and phone number. Also, please provide a list of all available transitaires. Begin!!!" Qin et al. (2023b)

## D  NUANCES COMPARING PRIOR WORK

Nuances comparing prior work from Table 5. ReAct and ART evaluate on hard QA tasks, but these tasks traditionally do not require the usage of tools to complete. We also note API-Bank fits all criteria. However, its level 1-2 examples are automated but simple. In comparison, its level 3 examples are complex but require manual evaluation. ToolBench's tasks require hard to use tool, but have shallow solution paths. Their more complex environments have deeper solution paths, but re-use the existing datasets of WebShop (Yao et al., 2022a) and TableTop (Liang et al., 2023).

