# OpenReview forum: "ToolTalk: Evaluating Tool Usage in a Conversational Setting"
_ICLR.cc/2024/Conference — Submitted to ICLR 2024_

### Official Review · Reviewer_NJBf · 2023-10-31

**Soundness:** 2 fair
**Presentation:** 2 fair
**Contribution:** 2 fair
**Rating:** 3
**Confidence:** 4

**Summary:**

This paper presents ToolTalk, a benchmark designed to evaluate the performance of language models in a conversational setting when using external tools to complete complex tasks. The benchmark includes a diverse set of user intents and external tools, and evaluates models based on their ability to complete tasks accurately and efficiently. The paper also includes an analysis of the performance of several state-of-the-art language models (GPT-3.5, GPT-4) on the ToolTalk benchmark, identifying areas where these models struggle and potential avenues for future research. Overall, ToolTalk provides a valuable resource for evaluating the performance of language models in a realistic conversational setting.

**Strengths:**

1. ToolTalk benchmark: The paper introduces ToolTalk, a benchmark consisting of complex user intents requiring multi-step tool usage specified through dialogue. ToolTalk contains 28 tools grouped into 7 plugins, and includes a complete simulated implementation of each tool, allowing for fully automated evaluation of assistants that rely on execution feedback. ToolTalk also emphasizes tools that externally affect the world rather than only tools for referencing or searching information.

2. Evaluation of language models: The paper evaluates the performance of several state-of-the-art language models on the ToolTalk benchmark, including GPT-3.5 and GPT-4. The analysis reveals areas where these models struggle, such as understanding complex user intents and effectively using external tools, and suggests potential avenues for future research.

**Weaknesses:**

1. The scope covered by the proposed benchmark is too narrow, compared to recent work on benchmarking the tool-using ability. Although I understand the authors may want to have a focus contribution to this specialized direction (a.k.a, conversation), the difference with other work is not that clear. For example, a lot of work recently tried to evaluate the tool-using ability to solve real-world tasks. See [1] for an example. In this case, real-world tasks can always happen during the conversation.

2. The evaluation is too limited. This is a fundamental drawback for this kind of benchmark paper. Basically, only two models, namely GPT-3.5-turbo and GPT-4, are considered in the experiments.  It would be more interesting to see the results of some open-sourced models, like Vicuna, and Alpaca.

3. The overall presentation needs to be improved. For example, the two algorithm blocks in Page-5 do not provide too much useful information.


[1] MINT: Evaluating LLMs in Multi-Turn Interaction with Tools and Language Feedback. Xingyao Wang, Zihan Wang, Jiateng Liu, Yangyi Chen, Lifan Yuan, Hao Peng, Heng Ji.

**Questions:**

None

---

> ### Author Response · Authors · 2023-11-20
>
> Thank you for your review and constructive feedback. See our response below:
>
> > The scope covered by the proposed benchmark is too narrow, compared to recent work on benchmarking the tool-using ability. Although I understand the authors may want to have a focus contribution to this specialized direction (a.k.a, conversation), the difference with other work is not that clear.
>
> As you pointed out, we focus on contributing to a specialized direction, conversation. We chose to do so since we are not trying to replace existing benchmarks for tool-using LLMs, but rather fill a gap apparent in existing work. Specifically, existing benchmarks fail to measure performance on complex tasks that require multiple utterances from a human to specify such as in a conversation. While this gap may seem narrow, it is of particular importance since we can expect most humans to interact with tool-using LLMs in this way, and indeed many already do such as ChatGPT, Anthropic's Claude, or Microsoft's Bing Chat.
>
> > For example, a lot of work recently tried to evaluate the tool-using ability to solve real-world tasks. See [1] for an example. In this case, real-world tasks can always happen during the conversation.
>
> While it is true real-world tasks often do occur during a conversation, a single real-world task does not necessarily encapsulate the entire goal of a user for the conversation. For example, [1] makes use of some popular mathematical reasoning datasets. We can imagine a scenario where a user wishes to answer a single math question in the conversation, in which case existing mathematical reasoning benchmarks might be sufficient. But we can also imagine scenarios where a user wishes to answer several, potentially dependent, math questions like a student, or even combine domains such as a user trying to balance dieting and a grocery budget. In this case, measuring performance on each individual instance outside of a conversation is less meaningful as failure to succeed at one or more instances would mean failure to accomplish the user’s end goal. Furthermore, specifying these user tasks in a conversation allows us to create complex tasks without having potentially long and unwieldy task descriptions. We find existing benchmarks do not capture these nuances of specifying tasks in a conversation resulting in the creation of ToolTalk.
>
> There are also several differences when comparing ToolTalk to [1] in particular. For one, [1] improves the realism of existing benchmarks by simulating a human in the loop when an incorrect answer is provided. However, the benchmarks re-used still specify their tasks in a single utterance, and thus the “dialogue” provided by [1] is not used to specify additional tasks in a conversation instead allowing for the LLM to “retry” at a single task. Thus, the concept and usage of dialogue is completely different between ToolTalk and [1].
>
> > The evaluation is too limited. This is a fundamental drawback for this kind of benchmark paper. Basically, only two models, namely GPT-3.5-turbo and GPT-4, are considered in the experiments. It would be more interesting to see the results of some open-sourced models, like Vicuna, and Alpaca.
>
> Thank you for your feedback, we were unable to evaluate more models due to time constraints. We are doing so for future revisions to our paper.
>
> > The overall presentation needs to be improved. For example, the two algorithm blocks in Page-5 do not provide too much useful information.
>
> Thank you for your feedback, what did you find confusing and additional information would be useful?
>
> ### References:
>
> [1] MINT: Evaluating LLMs in Multi-Turn Interaction with Tools and Language Feedback. Xingyao Wang, Zihan Wang, Jiateng Liu, Yangyi Chen, Lifan Yuan, Hao Peng, Heng Ji.

---

> ### Public Comment · ~Alexandru_Coca1 · 2024-04-25
>
> I feel that this review does not fairly assess the contribution of the authors. While, indeed, the proposed dataset is small and focused on a specific problem, such datasets are crucial for developing truly capable digital assistants and there is a conspicuous dearth of such data even among a flurry of benchmarks assessing LLM tool use capability. Meanwhile, citing [1] as an example of more comprehensive benchmark is rather unfair and misguided given its focus on arithmetic reasoning, code generation and embodied agents, tasks orthogonal to the settings considered in the present work. Even more, the contribution of [1] is focused on curating existing datasets as opposed to contributing new data.
>
> I also found points 2 and 3 overly critical. Additional evaluations on open source models are trivially achievable by the community. Having crisp and clear pseudo-code describing inference and evaluation procedures has definitely improved the reading experience and the clarity of the paper, for me.
>
> I hope the authors are not dissuaded by the negative tone of this review and they will focus on scaling their work which definitely provide significant benefit to research groups focused on grounded dialogue.

---

### Official Review · Reviewer_YHNY · 2023-11-01

**Soundness:** 4 excellent
**Presentation:** 3 good
**Contribution:** 3 good
**Rating:** 6
**Confidence:** 4

**Summary:**

This paper introduces a new benchmark, ToolTalk, to measure the performance of LLM-based assistants. It consists of complex user intents that require multi-step tool usage specified through dialogue. They evaluate GPT series models on ToolTalk and provide some insights for future research.

**Strengths:**

1. This paper proposes a new benchmark to evaluate the performance of LLM-based assistants. It includes a complete simulated implementation of each tool and emphasizes tools that externally affect the world. Their benchmark will contribute to the NLP community.

**Weaknesses:**

1. The number of involved tools is small, which may lead to biased evaluation for the LLMs measurement.
2. There is a lack of experiments on open-sourced LLMs, and how to enhance the ability of LLMs on their proposed benchmark is missing.

**Questions:**

None

---

> ### Author Response · Authors · 2023-11-21
>
> Thank you for your review and constructive feedback. See our response below:
>
> > The number of involved tools is small, which may lead to biased evaluation for the LLMs measurement.
>
> While the number of tools is small compared to recent works, we believe it is sufficient to show the shortcomings of GPT-4 and GPT-3.5. For example, both models tend to over-trigger, calling tools before all relevant information from the user has been supplied. This issue generally occurs for all tools though it is more consequential for action tools. While the occurrence of this problem might vary for any new, unseen tools, it seems safe to theorize that over-triggering is an issue not specific to any particular tool.
>
> We also generally chose the tools to reflect conversations we might expect for personal productivity, a topic that is likely to receive a lot of use for tool-augmented LLMs. So even in the unlikely event many of the issues we found in our analysis are only specific to these kinds of tools, we can expect many people to still experience these issues when conversing with a tool-augmented LLM.
>
> Ultimately, the current number of tools seems sufficient to demonstrate that evaluating tool-augmented LLMs in a conversational setting is a new task with unique characteristics that even some of the most state-of-the-art LLMs struggle to perform well on. However, adding additional tools is a clear target for future versions of ToolTalk,
>
> > There is a lack of experiments on open-sourced LLMs, and how to enhance the ability of LLMs on their proposed benchmark is missing.
>
> Thanks for your feedback. We plan on evaluating more LLMs for future revisions.

---

### Official Review · Reviewer_XpDT · 2023-11-01

**Soundness:** 2 fair
**Presentation:** 3 good
**Contribution:** 2 fair
**Rating:** 3
**Confidence:** 3

**Summary:**

This paper introduces ToolTalk, an evaluation benchmark for tool-usage in multi-turn dialogs.

There are 28 tools considered (belonging to 7 plugins such as Calendar/Email/Weather/etc), and each has (i) a high-level description, (ii) documentation about parameters, (iii) description of return value, (iv) simulated implementation in python, (v) equality metric to ascertain whether two invocations of a tool are equal. After the tools/plugins have been defined, (i) GPT4 is used to create complex scenarios requiring tool usage, (ii) humans (authors?) create tool-using conversations from these scenarios. Each turn in a multi-turn dialog consists of (i) user utterance, (ii) tool calls made by assistant, (iii) expected return values for tools, (iv) ground-truth assistant response. There are 50 conversations created with GPT4 scenarios (3 tool calls each), and 28 created by humans (one tool call).

The evaluation on ToolTalk is done in two phases: (1) Given a dialog ending in a user utterance, with all prior tool calls/responses, the model either (i) generates a response, or (ii) generates tool calls, which are executed until the model generates a response. (2) The predicted tool calls for a given context are compared against ground-truth tool calls. To measure the correctness of a tool call, either the arguments are compared for equality with the ground-truth (for action tools) or the results of the execution (e.g., search) are compared for equality (for non-action tools). This enables (1) precision of tool calls, (2) recall of tool calls and (3) success rate of tool calls (all tool calls correct, and no incorrect actions) and (4) incorrect action rate (wrong action tool is successfully called).

GPT-3.5 and GPT-4 are evaluated on ToolTalk, both with and without tool documentation. Three major reasons that models fail are: (i) premature tool calls, (ii) faulty reasoning and (iii) incorrect invocation of correct tool. For both with and without documentation, the rate of each error type is presented. The models without documentation perform worse.

**Strengths:**

The ToolTalk benchmark (data, metrics) can be a valuable resource to the research community. There are some important aspects that are carefully considered in the design of the data and the evaluation metrics, for example: (1) having a variety of realistic tools (action/non-action, different plugins, arguments), (2) having metrics for tool correctness and incorrect action rate.

**Weaknesses:**

- There is some lack of clarity about the creation of the ground-truth dialogs. It seems that it was written by humans, but how? Was there any validation of the data? Are the GPT4-generated scenarios biased in any particular way?

- It's not clear whether the user is always requesting that specific tools are invoked (e.g., "search my email", "check the weather") or whether there are situations wherein the assistant is expected to infer tool usage from a passive statement (e.g., "I forgot who emailed me about X" --> search_email(X), or "John asked me to do X by tomorrow" --> add_reminder(X, tomorrow)). Additional examples would help

- Will the proposed methodology generalize to a tool such a search engine or general-purpose retrieval? I imagine the tool correctness metric would need to change.

- The evaluation of GPT4/GPT3.5 does not provide much value, except to show that the benchmark is challenging and that tool documentation is important. To improve this, you could consider evaluating with alternate prompts (e.g., ones that induce better/worse tool usage) in order to demonstrate that the proposed benchmark can effective discern between tool-usage capabilities.

**Questions:**

see questions in weakness section

---

> ### Author Response · Authors · 2023-11-20
>
> Thank you for your review and constructive feedback. See our response below:
>
> > There is some lack of clarity about the creation of the ground-truth dialogs. It seems that it was written by humans, but how? Was there any validation of the data? Are the GPT4-generated scenarios biased in any particular way?
>
> We will add additional details about ToolTalk’s creation in the revision. To clarify now, all conversations were written by the authors using the scenarios as general guidelines. Step by step, our process involves:
>
> 1. Reading the scenario, discarding it if it is unrealistic, does not contain the required tools, does not involve a minimum of 3 tool calls, or is too similar to an already existing conversation.
> 2. Creating dialogue to match the scenario along with tool calls and arguments.
> 3. Update tool databases as appropriate to contain information needed for the newly created conversation.
> 4. Validate ground truth tool calls using an oracle predictor to ensure that ground truth tool calls and accompanying database changes are achievable.
> 5. Validate dialogue to ensure ground truth tool calls and arguments can be realistically inferred from the proceeding conversation without ambiguity. We do so by evaluating GPT4 on the conversation and check that failures are not due to bad dialogue.
>
> > It's not clear whether the user is always requesting that specific tools are invoked (e.g., "search my email", "check the weather") or whether there are situations wherein the assistant is expected to infer tool usage from a passive statement (e.g., "I forgot who emailed me about X" --> search_email(X), or "John asked me to do X by tomorrow" --> add_reminder(X, tomorrow)). Additional examples would help
>
> We plan on adding additional example visualizations in the revision. To clarify now, the assistant is generally expected to infer what tool to use. The creation of ToolTalk was done under the assumption that the user does not necessarily know the exact tools enabled by any particular plugin, and thus will not say things like “Use AddReminder to create a reminder to do X by tomorrow”.
>
> > Will the proposed methodology generalize to a tool such a search engine or general-purpose retrieval? I imagine the tool correctness metric would need to change.
>
> Yes, the methodology should extend to search. We already implement some basic search tools such as searching for emails or messages. These tools have a tool correctness metric that simply checks that all relevant emails or messages have been found regardless of the input to the tool. Given a static search database, a similar tool correctness metric could be implemented for search where we check for a few key search results and maybe additionally check that the search query is relatively similar to the ground truth.
>
> > The evaluation of GPT4/GPT3.5 does not provide much value, except to show that the benchmark is challenging and that tool documentation is important. To improve this, you could consider evaluating with alternate prompts (e.g., ones that induce better/worse tool usage) in order to demonstrate that the proposed benchmark can effectively discern between tool-usage capabilities.
>
> Thank you for your feedback, we plan on evaluating additional LLMs using ToolTalk in future revisions. We will also consider varying the prompts used as well.

---

### Official Review · Reviewer_LeeB · 2023-11-04

**Soundness:** 2 fair
**Presentation:** 3 good
**Contribution:** 2 fair
**Rating:** 3
**Confidence:** 3

**Summary:**

The paper presents a dataset that can evaluate tool calling in the context of a conversation. The dataset contains 28 tools from 7 categories. The approach to create conversations includes using GPT-4 with a set of tools (5) to accomplish a task. After a filtering process of these scenarios, 50 conversations are created manually with 28 more that are easy that includes one tool each.  The evaluation strategy includes fuzzy mechanisms to measure string based argument values, difference between actions and no-action tool calls etc. Their results and analyses shows that the success rates for the hard questions giving a good goal for the research community. The paper is written and presented well.

I have the following concerns with the overall work: (a) Dataset Size and Details: The dataset size seems very small 50 for hard and 28 for easy. Given that there are so much analyses done in the paper, it is concerning on it use as a good test set for conversational setting. Furthermore, it would be great to provide more details of the dataset after the filtering. How many turns in the conversations? How many average number of tools used and evaluated? etc. (b) GPT 3.5 and GPT 4: The only two models used for this work. It is important to know how other models such as Llama or Llama chat or Code Llama perform on this task. I do get the fact that GPT 4 would be the best in the performance but a comprehensive overview is important. (c) Evaluation metrics: Conversational systems are generally evaluated turn-wise with variations of rouge given the input from the APIs. But the overall results here just focus on the success rate and the toolcalls which raises the question on how useful the text and conversations are.

**Strengths:**

1. The work addresses an important topic in the context of conversational AI and Tools/APIs infusion in such a setting.
2. Datasets are few specifically those focused on sequences of tools or API calls in conversations.

**Weaknesses:**

1. Dataset size is very small and complete details are missing
2. Evaluation metrics only focus on tool calls and not conversations
3. Analyses is performed only on GPT3.5 and 4

**Questions:**

1. Is section 3.2 at the right place in evaluation methodology?
    2. Can you please specify the exact difference between easy and hard? Is it only one tool vs multiple tools?
    3. Can you please provide more statistics of the whole dataset after filtering? 50 conversations, how many turns, average number of tool calls?
    4. Did you manually evaluate the conversations for success rates? Is it possible that there were multiple plans for the same goal?
    5. Size of the dataset is a major concern for making any conclusions. Given that there are so much analyses already done on this dataset, how would you think it will help for evaluating as a test set — more seems like a development set.
    6. Clarify the exact differences between ToolLLM, ToolAlpaca, and APIBank. How does just the dialog setting enable research to develop new features?

---

### Meta-Review · Area_Chair_NpEr · 2023-12-06

**Metareview:**

This paper introduces ToolTalk, a benchmark dataset to evaluate LLMs on using tools in a conversational setting. ToolTalk features 78 user-agent dialogs across tools in domains such as accounting, email and calendar management. Evaluation on GPT models shows that while they achieve strong task success rates (85%-92%) on the 28 easy dialogs, they struggle on solving the remaining 50 challenging dialogs (26%-50%), suggesting future directions for improvement.

**Strengths**

* All the reviewers agreed that ToolTalk could be a useful resource to the research community as it “addresses an important topic in the context of conversational AI and tool-use” (LeeB) and comes with a sandbox environment to simulate the implementation of each tool (NJBf, YHNY, XpDT).

* Evaluation on strong GPT models suggests opportunities to further improve existing chat bots in interactive tool use (NJBf)

The design of evaluation metrics is also a contribution besides the data artifact (XpDT)

**Weaknesses**

* A major issue is the size of the dataset, which only comes with 78 dialogues (50 hard and 28 easy), as flagged by LeeB. Such a scarcity of evaluation dialogues could make it difficult to obtain statically robust results, and there is no discussion about variance in experiments in the paper.

* Analysis is limited to GPT models without using any open models (LeeB, YHNY, NJBf).

* The coverage of tools might be limited as compared to existing benchmarks (NJBf). AC: this is not considered as an issue in the decision since the other paper came out shortly before the submission deadline and it’s not been peer-reviewed yet.

* Lack of clarity in the creation of evaluation dialogues, such as the style of the natural language utterances (XpDT).

Given that this is a benchmark paper and the issues with the dataset size and the limitation in the models used, the decision is "Reject".

**Justification For Why Not Higher Score:**

The primary reason for not accepting this paper is that the benchmark dataset is too small (78 dialogues only) and there is no analysis on the variance in the experimental results.

**Justification For Why Not Lower Score:**

N/A

---

### Decision · Program_Chairs · 2024-01-16

Reject